# Screening of *Staphylococcus aureus* for Disinfection Evaluation and Transcriptome Analysis of High Tolerance to Chlorine-Containing Disinfectants

**DOI:** 10.3390/microorganisms11020475

**Published:** 2023-02-14

**Authors:** Yixiao Li, Yang Song, Zhenzhou Huang, Li Mei, Mengnan Jiang, Duochun Wang, Qiang Wei

**Affiliations:** 1National Pathogen Resource Center, Chinese Center for Disease Control and Prevention (China CDC), Beijing 102206, China; 2Division of Infectious Disease, Key Laboratory of Surveillance and Early-Warning on Infectious Disease, Chinese Center for Disease Control and Prevention, Beijing 102206, China; 3National Institute for Communicable Disease Control and Prevention, Chinese Center for Disease Control and Prevention, Beijing 102206, China

**Keywords:** standard strain, *Staphylococcus aureus*, MIC, tolerance to chlorine-containing disinfectants, DEGs

## Abstract

The nonstandard use of disinfectants can lead to the disinfectant resistance of bacteria and even increase antibiotic resistance. However, compared with the study of antibiotic resistance, studies of bacterial resistance to disinfectants are relatively few in number. In this study, we explored the standard strain screening procedure for the evaluation of disinfection efficacy. *Staphylococcus aureus* strains with different sources and substrates were selected from the National Pathogen Resource Center of China and screened the standard strains that could evaluate the long-term bacteriostatic effect of the chlorine-containing disinfectants through the determination of the physical properties, genome-based safety evaluation, and disinfection test evaluation. In this process, one *S. aureus* strain was more resistant to the long-term bacteriostasis of chlorine-containing disinfectants than the other strains. This strain and the standard strain ATCC 6538 were cultured in the medium containing a low concentration of chlorine-containing disinfectant synchronously. Then, comparative transcriptome analysis was carried out to investigate the potential mechanism of a high tolerance to chlorine-containing disinfectants. The pathway of significant differential expression is related to the oxocarboxylic acid metabolic mechanism, amino acid metabolic mechanism, and pyrimidine mechanism, which may be the molecular mechanism of *S. aureus* evolution to adapt to chlorine-containing disinfectants. Our study established a technical process for screening and evaluating standard strains for disinfection, which also provided a reference for studying the bacterial evolution mechanism toward chlorine tolerance.

## 1. Introduction

Infectious diseases can form pandemics in a short time, cause public health events, and seriously endanger public health. Preventing infectious diseases is a global challenge [1]. Disinfection treatment time and effectiveness play vital roles in preventing and controlling the epidemic. The most common measure of disinfection is chemical disinfection. Chemical disinfectants act on lifeless surfaces to quickly kill or inactivate microorganisms [2,3]. The testing of disinfectants is needed for safety, efficacy, and quality control [3]. Disinfection standard strains are used for disinfectant evaluation. *Staphylococcus aureus* (*S. aureus*) is a critical pathogen in health care-associated infections (HAIs) and food-borne infections [4,5]. ATCC 6538 is a standard testing strain for disinfectants [6,7].

During the COVID-19 pandemic, disinfectants, alcohol-based hand sanitizers, and antiseptic hand washing surged [8]. As a preventive measure, many authorities have also increased the chlorine dosage in wastewater disinfection, far exceeding the prescribed dosage [9]. Just as bacteria develop antibiotic resistance after the action of antibiotics, the nonstandard use of chemical disinfectants will also enhance the adaptability of pathogenic bacteria to the environment. The surviving bacteria, after disinfection, will have physiological changes and may regenerate [10]. In this way, bacteria are resistant to disinfectants. In the long run, bacterial resistance to disinfectants may increase. In addition, the abuse of disinfectants will not only cause an increase in bacterial resistance to disinfectants, but also lead to an increase in bacterial antibiotic resistance. For example, the disinfection by-products (DBPs) chlorite and iodoacetic acid in water have antibiotic-like effects, which will lead to the evolution of *Escherichia coli* drug resistance [11]. Widespread use of benzalkonium chloride disinfectant will promote the development of *Pseudomonas aeruginosa* drug resistance [12]. Through co-resistance and cross-resistance, strains resistant to low concentrations of disinfectants may also develop drug resistance [13]. In this case, whether using the previous standard strains could still achieve the objective of evaluation is unclear. A series of evaluation methods need to be developed to eliminate the influence of interference factors such as the temperature, pH, and organic content [14] and ensure that the strain can achieve the purpose of the evaluation of disinfectants. At the same time, it also helps to cope with the possible trend of increasing bacterial resistance.

In recent years, the research on the effect of disinfectants on bacteria at the genomic and transcriptome levels is gradually increasing. Presently, the known mechanisms for the formation of disinfectant resistance include the acquisition of mobile genetic elements, phenotypic changes, drug effect pumps, changes in antibacterial targets, inactivated disinfectants, etc. [15]. Destruction of the cell plasma membrane, protein degradation, and DNA damage may be the way for disinfectants to inactivate bacteria [16,17]. Studies on the inactivation of bacteria by chlorine-containing disinfectants have indicated that bacteria might be inactivated by the oxidative damage of cell components [18]. There are a few reports on the mechanism of high bacterial tolerance to disinfectants.

This study used the microbial resources in the National Pathogen Resource Center (NPRC) of China to screen standard strains for the evaluation of disinfection efficacy. While screening standard strains for disinfection, we found a strain of *S. aureus* CHPC 1.8487 that had neither the penam antibiotic gene *mecA* nor the anti-disinfectant gene *qac* families. Compared with the disinfection standard strain ATCC 6538, it had higher resistance to chlorine disinfectant in the presence of organic matter. Therefore, we analyzed the differential expression gene (DEG) profiles between this strain and ATCC 6538 at the transcriptome level after culture in the medium containing a low chlorine concentration to further understand the molecular mechanism of the high tolerance of *S. aureus* to chlorine-containing disinfectants.

## 2. Materials and Methods

### 2.1. Screening of Standard Strains for Long-Term Bacteriostatic Efficacy Evaluation of Chlorine-Containing Disinfectants

#### 2.1.1. Strain Selection and Bacterial Suspension Preparation

In NPRC, we selected *S. aureus* strains from different regions and substrates. After morphological identification, biochemical identification, and 16S rRNA identification, 13 strains of *S. aureus* were selected as candidate strains. The collection numbers are CHP 1.8391, CHPC 1.8395, CHPC 1.8399, CHPC 1.8427, CHPC 1.8431, CHPC 1.8485, CHPC 1.8486, CHPC 1.8487, CHPC 1.8492, CHPC 1.8495, CHPC 1.8498, CHPC 1.8507, and CHPC 1.8601. In subsequent experiments, these strains were tested together with standard ATCC 6538.

Glycerol tubes were taken from the −80 °C refrigerator and inoculated in nutrient agar (NA, Hopebio, Qingdao, China) after thawing at room temperature and incubated at 37 °C for 24 h. Next, we picked up the monoclonal strain for passage and take the third-generation new culture (18–24 h) into the tryptone physiological solution (TPS, OXOID, Beijing, China). In this way, preliminarily, we prepared the bacterial suspension.

#### 2.1.2. Acid Alkali Resistance Test and Heat Resistance Test

The acid test range of pH was between 2 and 4. The alkali test pH range was between 10 and 12 [19]. These pH amounts were adjusted by adding sodium hydroxide (Merck, Darmstadt, Germany) and hydrochloric acid (Merck, Germany) to the trypticase soy broth (TSB, OXOID, Beijing, China) [20] and took pH = 7 as the control. The pH was controlled and adjusted with a digital pH meter (PHS-3CT, Kangyi, Shanghai, China). The temperature test range of the culture temperature was 45 °C and 50 °C [21], and we took 37 °C as the control. We adjusted the active concentration of the bacterial suspension to 1 × 10^5^ cfu/mL~5 × 10^5^ cfu/mL. After 24 h of culture, counted the living bacteria by tryptone soy agar (TSA, OXOID, Beijing, China) plate. Next, we recorded the number of bacterial colonies (cfu/mL), repeated the test three times, and took the average value as the final number of living bacteria. Finally, we converted the number into a logarithmic value.

#### 2.1.3. Genome Sequencing, Antimicrobial Resistance Genes Analysis, and Virulence-Related Genes Analysis

The genomes of the standard strain and candidate strains were extracted by the Wizard Genomic DNA Extraction Kit (Promega, Madison, WI, USA) following the manufacturer’s instructions. The HiSeq sequencer (Illumina HiSeq2000, San Diego, CA, USA) was used to perform 500-bp paired-end whole-genome sequencing with 290× coverage. FastQC (https://www.bioinformatics.babraham.ac.uk/projects/fastqc/ accessed on 18 July 2022.) was used to evaluate the quality of the reads (low-quality reads were discarded if the quality scores of ≥3 consecutive bases were ≤Q30). Readfq (version 10) was used to filter the original data to obtain adequate data (clean data). SOAP de novo (Version 2.04) was used to assemble the clean data of each strain and finally integrate it with CISA (Contig Integrator for Sequence Assembly) [22,23].

The Comprehensive Antibiotic Research Database (CARD, http://arpcard.mcmaster.ca accessed on 1 August 2022) was used to predict the potential antibiotic resistance genes of these strains [22]. The parameters of BLAST+ were a percentage of sequence identity ≥ 98% and a percentage of length coverage ≥ 98%. The Virulence Factor Database (VFDB, http://www.mgc.ac.cn/VFs/ accessed on 1 August 2022) was used to predict the virulence-related genes of these strains.

#### 2.1.4. Disinfection Test

##### Neutralizer Identification Test

We prepared a disinfectant solution with an effective chlorine content of 500 mg/L using distilled water and 84 disinfectants (LIRCON, Dezhou, China). The neutralizer was a 5 g/L sodium thiosulfate solution (XiLONG SCIENTIFIC, Shantou, China). The reference strain was ATCC 6538. We set up six groups to confirm that the neutralizer can stop the action of residual disinfectant according to *EN 1276* and the *Technical Standard for Disinfection* of China [24,25,26]. In the group 1 tube, we added the disinfectant and bacterial suspension. In the group 2 tubes, we added the disinfectant and bacterial suspension into the neutralizer. In the group 3 tubes, we added a neutralizer and bacterial suspension. In the group 4 tubes, we added the product of neutralization and bacterial suspension. In the group 5 tubes, we added TPS and bacterial suspension; in the group 6 tubes, we added TPS, a neutralizer, and culture medium. Finally, we counted the living bacteria after incubation at 37 °C for 48 h.

##### Minimum Inhibitory Concentration (MIC) Test and Minimal Bactericide Concentration (MBC) Test

The MIC and MBC tests were carried out on the remaining strains of *S. aureus*. We used distilled water to double dilute 84 disinfectants into test solutions of different concentrations (2000 mg/L, 1000 mg/L, 500 mg/L). As per the *Technical Standard for Disinfection* [26] of China, we took 2.5 mL of the test solutions of various dilutions and added it into the test tube containing 2.5 mL of nutrient broth (NB, Hopebio, Qingdao, China) of double concentration. We took 0.1 mL of bacterial suspension with a bacterial content of about 10^8^ cfu/mL into the test tube of NB containing disinfectant as the sample of the experimental group before inoculating the bacteria into the NB tube without disinfectant as the positive control sample. The actual action concentration was 5 × 10^5^~5 × 10^6^ cfu/mL. We took two NB tubes as samples of the negative control group, put them into the incubator at 37 °C for 48 h, repeated the experiment three times, and observed the results. The experimental results showed that the MIC values of the standard and candidate strains were 500 mg/L. To narrow the screening range, we set 300 mg/L and 400 mg/L as the chlorine concentrations for the MIC experiments, and repeated the experiment six times.

When the positive control tube was visibly turbid, the negative control tube was clear. Therefore, the MIC value is the disinfectant concentration corresponding to the highest dilution of the test group [26].

We then took 0.5 mL of the clarification liquid into 1.5 mL of neutralizer and mixed them well. After 10 min of action, it was inoculated into the nutrient agar (NA) plate and cultivated at 37 °C for 48 h. The MBC value is the minimum disinfectant concentration without colony growth [13].

##### Quantitative Germicidal Test

The quantitative germicidal test is the most important link in the laboratory test to evaluate the effect of the disinfectants, and the test concentration was the minimum concentration of disinfectants [26]. According to *EN 1276*, we diluted the disinfectant with hard water to 1.25 times the experimental concentration, and the experimental concentration was 250 mg/L available chlorine. The initial bacterial suspension concentration was adjusted to 1 × 10^8^ cfu/mL~5 × 10^8^ cfu/mL, then 0.5 mL of the test suspension and 0.5 mL of organic interfering substance were added into the test tube. We mixed 4.0 mL of the disinfectant, started the stopwatch immediately, and placed the tube in a water bath controlled at 20 ± 1 °C for 10 min and 30 min. At the end of the test time, 0.5 mL of the mixed solution was added to 4.5 mL of neutralizer for 10 min. Then, each tube was inoculated into two TSA plates to count the number of viable bacteria. We used the diluent instead of disinfectant to conduct the parallel experiment as the positive control. All samples were incubated in a 37 °C incubator for 48 h [24,26,27]. The experiment was repeated three times and then the concentration of viable bacteria in each group (cfu/mL) was calculated and converted it into a logarithm. The logarithmic value of the average viable bacteria concentration in the control group was denoted by N_0_. The logarithmic value of the average viable bacteria concentration in the test group was denoted by N_x_. The killing logarithm (KL) was calculated according to the following formula of KL = N_0_ − N_x_. If the average number of the colony after disinfection treatment was less than 1, the killing logarithm was recorded as “>7”, which was not included in the statistical analysis.

### 2.2. Differentially Expressed Genes (DEGs) Analysis Intolerance to Chlorine-Containing Disinfectants

Strains were selected with high tolerance to chlorine-containing disinfectants in the MIC experiment, that is, high MIC value. Chlorine-containing disinfectants were added to the NB medium to produce a final chlorine concentration of 250 mg/L. The highly tolerant strain and ATCC 6538 were cultured in this medium synchronously. These two strains were cultured in the NB medium without disinfectant. All samples were incubated in a 37 °C incubator for 24 h. Then, the four samples were centrifuged and the supernatant discarded. These were then washed with distilled water to remove the disinfectant and culture solution. The total RNA of the four samples was extracted by the RNeasy Mini Kit (Qiagen, Hilden, Germany) following the manufacturer’s instructions. The cDNA was reverse transcribed from total RNA by HiFiScript gDNA Removal RT MasterMix (CWBIO, China) and submitted to Novogene (Beijing, China) for transcriptome database building and sequencing. The differentially expressed genes were annotated against the Gene Ontology (GO) and Kyoto Encyclopedia of Genes and Genomes (KEGG) databases. The GO term functional analysis and KEGG pathway enrichment analysis were performed in subsequent analyses [28]. In this way, we further understood the molecular mechanism of the high tolerance of *S. aureus* to chlorine-containing disinfectants.

## 3. Results

### 3.1. Screening of Standard Strains for Long-Term Bacteriostatic Efficacy Evaluation of Chlorine-Containing Disinfectants

#### 3.1.1. Comparison of Acid–Alkali Resistance and Heat Resistance

All candidate strains were consistent with the standard ATCC 6538 in terms of acid, alkaline, and heat resistance. The logarithms of viable bacteria concentration under different culture conditions are shown in Figure 1. All strains could survive in the TSB medium with pH = 4 and pH = 10 but could not survive in the TSB medium with pH = 3 and pH = 11. Moreover, they could survive at 45 °C but not at 50 °C.

#### 3.1.2. Genome Analysis

Twenty-six drug-resistance genes were predicted in the 14 *S. aureus* strains (Appendix A). The comparison showed that all strains did not contain the *qac* family, a kind of quaternary ammonium disinfectant resistance gene. As shown in Figure 2, the different drug resistance genes among strains were macrolide antibiotic genes (including *ErmA*, *ErmB*, and *ErmC*), tetracycline antibiotic gene tetK, fosfomycin antibiotic genes (including *FosB*, *GlpT*, and *murA*), penam antibiotic genes (including *mecA* and *mecR1*), aminoglycoside antibiotic genes including *AAC(6′)-le-APH(2″)-la, ANT(4′)-lb* and *ANT(9)-la*, rifamycin antibiotic gene rpoB, and fluoroquinolone antibiotic genes (including *norA*, *parC*, *parE*, *gyrA*, and *sdrM*).

The common drug-resistant bacteria of *S. aureus* is methicillin-resistant *Staphylococcus aureus* (MRSA). Infections due to MRSA are associated with higher mortality rates than infections caused by methicillin-susceptible strains [29]. MRSA strains produce an altered penicillin-binding protein (PBP), and PBP can produce a protein with a low binding capacity of β-lactam antibiotics. The protein is encoded by an acquired gene, *mecA* [30]. Therefore, MRSA can grow in the presence of β-lactam antibiotics. Therefore, in the screening process step, CHPC 1.8498, CHPC 1.8507, and CHPC 1.8601 containing penam antibiotic genes were excluded.

Twenty-nine virulence-related genes were predicted in the 14 *S. aureus* strains. These virulence-related genes can be clustered into five groups (Table 1). All strains have the following virulence factors: Adherence-related genes (including *atl*, *ebp*, *efb*, *fnbA*, *fnbB*, *icaA*, *sdrC*, and *spa*), enzyme-related genes (including *sspA*, *sspB*, *sspC*, *hysA*, *geh*, *lip*, *coa*, *sak*, and *nuc*), immune evasion-related gene *sbi*, secretion system-related genes (including *esaA*, *essA*, and *esxA*), and toxin-related genes (including *hly*, *hla*, *hld*, *hlgA*, and *hlgB*). The different distribution of virulence factors of 14 strains is shown in Figure 2. In the differential distribution of enterotoxin and enterotoxin-like virulence factors, ATCC 6538 contained enterotoxin-like genes *selk* and *selq*, which did not contain the enterotoxin gene. CHPC 1.8391, CHPC 1.8399, CHPC 1.8427, CHPC 1.8431, CHPC 1.8487, CHPC 1.8492, CHPC 1.8495, and CHPC 1.8507 contained enterotoxin genes *sea* or *seb*. CHPC 1.8395, CHPC 1.8485, CHPC 1.8486, and CHPC 1.8601 did not contain enterotoxin and enterotoxin-like genes. In the differential distribution of panton-valentine leukocidin (PVL) virulence factors, ATCC 6538 contained the *lukD* gene; CHPC 1.8485, and CHPC 1.8601 did not contain the PVL virulence-related gene.

Staphylococcal enterotoxins (SEs) are subdivided into classical SEs with emetic activity and SE-like proteins with no or relatively low emetic activity. SEs play an essential role in staphylococcal food poisoning, and *seb* can inhibit keratinocyte proliferation and migration and may delay wound closure [31]. Panton-valentine leukocidin (PVL) is an extracellular protein with dermonecrotic and leucocidal functions. In addition, it has cytotoxic activity against mammalian neutrophils, monocytes, and macrophages [31]. Combined with the distribution of the drug-resistance gene and virulence-related gene, the safety at the gene level of CHPC 1.8395, CHPC 1.8485, and CHPC 1.8486 was similar or higher than that of ATCC 6538.

#### 3.1.3. Disinfection Test Result

##### Neutralizer Test Result

The neutralizer qualified if the following conditions were satisfied. Group 1 was sterile, or only a few colonies grew. This group was used to observe whether the disinfectant had the ability to kill or inhibit the test bacteria. The number of colonies in group 2 was more than that in group 1, but less than that in groups 3, 4, or 5. Group 2 was used to observe whether the test bacteria treated by the disinfectant could resume after the residual disinfectant was neutralized. If group 1 is sterile, the count of group 2 will be >5. If the count of group 1 is between 1 and 10 (denoted as X), group 2 shall be >(X + 5). If the count of group 1 is more than 10 (denoted as Y), group 2 shall be >(Y + 0.5Y). Group 3 was used to observe whether the neutralizer was bacteriostatic, and group 4 was used to observe whether the neutralized product or the residual disinfectant that had not been completely neutralized had any effect on the growth and reproduction of the test bacteria, and the group 5 was used as the bacterial count control. Groups 3, 4, and 5 had similar bacterial growth, and the colony error rate between groups was less than 15%. Group 6 was used as negative control for the sterile groups.

The neutralizer test results (in Table 2) show that groups 1, 2, and 6 met the requirements. In addition, the error rate between groups 3, 4, and 5 was less than 15% each time. The results show that neutralizers can neutralize the influence of disinfectants without inhibiting bacterial growth.

##### MIC and MBC Results

After screening of the acid and alkali resistance test, heat resistance test, drug resistance gene, and disinfectant resistance gene, there were 10 strains remaining. These 10 strains and ATCC 6538 were subjected to subsequent disinfection experiments. At first, when the experimental concentration of the chlorine-containing disinfectant was 1000 mg/L, 500 mg/L, and 250 mg/L, the MIC value and MBC value of all strains were 500 mg/L. At this time, all candidate strains had the same long-term bacteriostatic effect on the chlorine-containing disinfectant. Later, we wanted to explore the difference in each candidate strain, so we narrowed the range of disinfectant concentration, set as 300 mg/L and 400 mg/L, and calculated the trimmed mean through multiple experiments. As shown in Figure 3, except for CHPC 1.8391 and CHPC 1.8487, the MIC value and MBC value of the other strains were consistent with ATCC 6538. The MIC value and MBC value of CHPC1.8487 were relatively high. From the results, this strain has higher resistance to a chlorine disinfectant than other strains in the presence of organic matter.

##### Quantitative Germicidal Test Result

The quantitative germicidal test results of 250 mg/L available chlorine for 10 min and 30 min are shown in Table 3. Except for CHPC 1.8485 and CHPC 1.8495, all candidate strains and standard strains survived at 10 min. After the 30 min reactions, all strains could not survive. After removing CHPC 1.8485 and CHPC 1.8495, the KL value after the 10 min reaction was statistically tested. We took the KL value of the standard strain ATCC 6538 as a parameter, and calculated the 95% confidence interval according to the formula $\left(KL\pm t_{\alpha /2,\;\nu }\frac{s}{\sqrt{n}}\right)$. The result was (3.868, 4.876). The strains in this range were CHPC 1.8391 and CHPC 1.8395.

Based on the above results, in terms of the long-term bacteriostatic effect on chlorine-containing disinfectants, CHPC 1.8395 had the same effect as ATCC 6538. Therefore, CHPC 1.8395 can be considered as a standard strain for testing the long-term bacteriostatic effect of chlorine-containing disinfectants. At the same time, we found that strain CHPC 1.8487 had a higher chlorine tolerance in the MIC test. This strain had a low KL value, so its resistance to the instantaneous killing of chlorine-containing disinfectants was weak. However, the tolerance to chlorine-containing disinfectants was higher in the presence of nutrients.

### 3.2. DEGs of Resistance to Chlorine-Containing Disinfectants between S. aureus ATCC6538 and CHPC 1.8487

#### 3.2.1. Gene Expression Analysis

After culturing the two strains in NB medium containing 250 mg/L chlorine for 24 h, comparative transcriptome analysis was conducted to discover genes involved in the tolerance to chlorine-containing disinfectants. The box plot of the total gene expression distribution (Figure 4a) showed the distribution of the total gene expression levels of ATCC 6538 and CHPC 1.8487 under two conditions. Comparing the relative transcript abundance in each gene using the FPKM, in ATCC 6538, a total of 534 genes were found to be differentially expressed (Figure 4b), 245 of these were significantly upregulated, whereas 190 genes were significantly downregulated. In CHPC 1.8487, 290 genes were found to be differentially expressed (Figure 4c); 153 were significantly upregulated, whereas 137 genes were significantly downregulated.

#### 3.2.2. GO Enrichment and KEGG Pathway Analysis

The GO annotation analysis (Figure 5) categorized the DEGs into three modules: biological process (BP), cellular component (CC), and molecular function (MF). In ATCC 6538, the top five GO terms mainly relevant for the upregulated DEGs contained “membrane (GO:0016020)”, “establishment of localization (GO:0051234)”, “localization (GO:0051179)”, “transport (GO:0006810)”, and “small molecule binding (GO:0036094)”. The downregulated unigene group was the same as the upregulated unigene group. In CHPC 1.8487, the top five GO terms mainly relevant for the upregulated DEGs contained “membrane (GO:0016020)”, “establishment of localization (GO:0051234)”, “localization (GO:0051179)”, “transport (GO:0006810)”, and “transporter activity (GO:0005215)”. The downregulated unigene group was the same as the upregulated unigene group. The common unigene group in the two strains that play a role in the resistance of chlorine-containing disinfectants were “membrane”, “establishment of localization”, “localization”, and “transport.” “Transportation activity” might play a more significant role in the tolerance of chlorine-containing disinfectants.

To further comprehensively understand the enriched metabolic or signal transduction pathways, we classified the DEGs in the KEGG pathway database for enrichment analysis. The top 20 pathways related to the differentially expressed genes after treatment with chlorine-containing disinfectants are depicted in Figure 6. As a result, “phosphotransferase system (PTS)”, “ABC transporters”, and “*Staphylococcus aureus* infection” had a significant difference in the enrichment analysis of ATCC 6538 (adjusted *p* value < 0.05). Furthermore, “2-oxocarboxylic acid metabolism”, “biosynthesis of amino acids”, “*Staphylococcus aureus* infection”, “valine, leucine, and isoleucine biosynthesis”, “pyrimidine metabolism”, and “ABC transporters” had a significant difference in the enrichment analysis of CHPC 1.8487 (adjusted *p* value < 0.05). The common pathway for the two strains were “ABC transporters”, “*Staphylococcus aureus* infection”, and “*Staphylococcus aureus* infection”. The CHPC 1.8487 unique pathway was “2-oxocarboxylic acid metabolism”, “biosynthesis of amino acids”, “valine, leucine, and isoleucine biosynthesis”, and “pyrimidine metabolism”. Moreover, these unique pathways may play a more significant role in the tolerance to chlorine-containing disinfectants. The cluster analysis results of the above four pathways genes of CHPC 1.8487 are shown in Figure 7 as *tpiA*, *hisB*, and *carB* were upregulated, which regulate triosephosphate isomerase, histidine metabolism, and carbamoyl phosphate synthase.

## 4. Discussion

Disinfection standard strains are available microbial resources. In addition to disinfection, pathogenic microorganisms are valuable biological resources in biosafety, human health, environmental protection, and renewable energy [32]. Screening standard strains for disinfection from NPRC is a new way to use the Pathogen Microbe Resource Center fully. This approach opens new ideas for the utilization of pathogenic resource centers. Moreover, the resource center provides a basis for research in various fields [33,34].

In this study, we put forward a technical screening process for disinfection standard strains and an evaluation method for candidate strains. First, the effects of temperature, pH, and organic matter should be excluded as the temperature will affect the composition of the extracellular vesicles (EVs) of *S. aureus*, which is related to cytotoxicity [35]. The pH is effective in biofilm formation [36] as the ability to form an *S. aureus* biofilm is strongly affected by the pH [20,37]. Moreover, alkaline pH can control the biofilm structure, so the biofilm becomes much weaker and thinner [38]. The biofilm matrix presents a diffusion barrier and a neutralizing environment for some biocides [3] to protect bacteria. At the same time, they use the medium containing organic substances for disinfection tests to ensure that these factors have the same impact on the results in the subsequent evaluation of candidate strains by comparing their drug-resistance genes and virulence-related genes. The strains with β-lactam antibiotics-related genes were removed to reduce the interference of drug resistance on the disinfection effect at the gene level. Considering biological safety, this step reduces the screened strain’s potential pathogenicity [39]. This technical process can provide a reference for the future safety evaluation of standard strains. Then, the standard strain of *S. aureus* ATCC 6538 for disinfection was used to test the chlorine-containing disinfectants and their long-term antibacterial ability. We finally screened out *S. aureus* strain CHPC 1.8395 (CGMCC 25904, patent application No. 202211485300.4), which can evaluate the long-term antibacterial ability of chlorine-containing disinfectants.

In this screening process, we found a strain CHPC 1.8487 with higher tolerance to the long-term bacteriostasis of chlorine-containing disinfectants than the disinfection standard strain ATCC 6538. In the MIC test, the result that was obtained by the double dilution method clearly shows that the experimental strains could not survive at 500 mg/L effective chlorine but could survive at 250 mg/L. In the process of reducing the range of MIC values, we found that the MIC value was not always limited to a small range. Therefore, we repeated the experiment and obtained the results of this study through the trimmed mean. Base on this premise, CHPC 1.8487 had a higher chlorine resistance than other strains. We took this as the research object for preliminary exploration and analysis. Through comparative transcriptome analysis, we found that the KEGG pathways related to tolerance to chlorine-containing disinfectants were oxocarboxylic acid metabolism, amino acid metabolic metabolism, and pyrimidine metabolic metabolism. This study showed that the tolerance responses of *S. aureus* to chlorine-containing disinfectants are mediated by complex molecular and metabolic-related responses.

Long-term exposure to biocides induced biofilm formation and shut down motility [40]. Furthermore, chlorine has been reported to impact the metabolic processes, efux pumps, and pathogenicity of bacteria by regulating gene expression [41]. In this study, expression changes of “membrane” and “ABC transporters” at the transcriptome level also confirmed that the tolerance of *S. aureus* to chlorine-containing disinfectants works through the biofilm and efflux pump.

In past studies, regulating the triosephosphate isomerase gene *tpiA* was related to bacterial resistance to aminoglycoside antibiotics [42]. In this study, the expression of *tpiA* in the higher tolerance strain was upregulated, indicating that *tpiA* may be related to the high tolerance of *S. aureus* to chlorine-containing disinfectants. It also suggests that *tpiA* may be related to the cross-resistance between disinfectants and antibiotics. Furthermore, histidine starvation conditions increased the *hisB* transcript level, and deletion of the complete hisB open reading frame resulted in histidine auxotrophy in Aspergillus nidulans [43]. In this study, the upregulated expression of *hisB* suggests that increasing histidine metabolism may enhance bacterial tolerance to chlorine. In addition, *carB* can affect the nematicidal activity by modulating the pH environment [44]. In our study, *carB* was upregulated, suggesting that *carB* may be related to cross-resistance between nematicides and chlorine-containing disinfectants.

In the fight against infectious diseases, antibiotics and disinfectants have become powerful weapons for humans. However, both antibiotics and disinfectants are harsh environments for microorganisms. Microorganisms have developed various strategies to fill their niche and survive adversity [28]. There are many studies on horizontal gene transfer including the transfer of mobile genetic elements such as a plasmids, phages, and integrons [45]. Therefore, it is an important way for microorganisms to adapt to the environment and accelerate evolution. In routine disinfection, disinfectants will remain in the environment, stimulating microorganisms to respond to environmental changes. The stress of disinfectants has caused microorganisms to form various biological mechanisms and physiological reactions, which can help them survive in an adverse environment. When the residues of chlorine-containing disinfectants in the environment cannot reach the concentration to eliminate bacteria, *S. aureus* will change its mechanism to adapt to this stimulus. The oxocarboxylic acid metabolic mechanism, amino acid metabolic mechanism, and pyrimidine mechanism may be the molecular mechanism of *S. aureus* evolution to adapt to chlorine-containing disinfectants.

## 5. Conclusions

In order to cope with the possible trend of the increase in bacterial disinfectant resistance and make full use of the pathogenic microbe resource center, we explored and established a set of disinfection standard strain screening technical processes. Furthermore, an *S. aureus* strain can be used to evaluate the chlorine-containing disinfectants and long-term bacteriostatic ability. This technical process can provide a reference for future research on standard strains. This practice also provides a new model for using a pathogenic microbe resource center. At the same time, we found an *S. aureus* strain with high tolerance to chlorine. At the transcriptome level, four pathways with high tolerance to chlorine were found, namely, “2-oxocarboxylic acid metabolism”, “biosynthesis of amino acids,” “valine, leucine, and isoleucine biosynthesis”, and “pyrimidine metabolism”. It was also suggested that the bacterial cross-resistance mechanism with disinfectants and antibiotics and the cross-resistance mechanism with disinfectants and nematicides provide a reference for the evolution mechanism research of bacterial chlorine tolerance.

## 6. Patents

We finally screened out an S. aureus strain CHPC 1.8395 (CGMCC 25904, patent application No. 202211485300.4) that can evaluate the long-term antibacterial ability of chlorine-containing disinfectants.

## Figures and Tables

**Figure 1 microorganisms-11-00475-f001:**
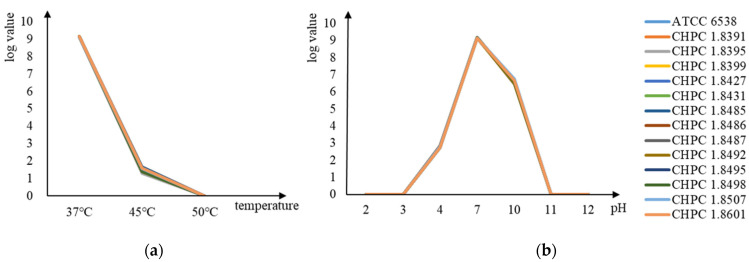
Growth condition of 14 strains in different environments. (**a**) The logarithm of viable bacteria of 14 strains under different pH conditions. (**b**) The logarithmic number of viable bacteria of 14 strains under different temperature conditions.

**Figure 2 microorganisms-11-00475-f002:**
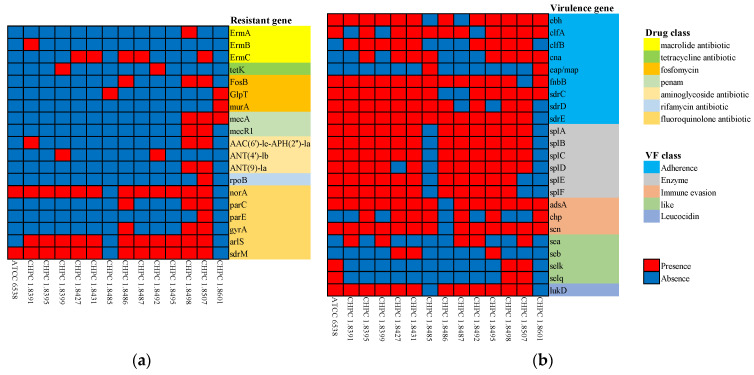
Differential distribution heat map of the drug resistance and virulence factors of 14 strains. (**a**) Differential distribution heat map of the drug resistance of 14 strains. (**b**) Differential distribution heat map of the virulence factors of 14 strains.

**Figure 3 microorganisms-11-00475-f003:**
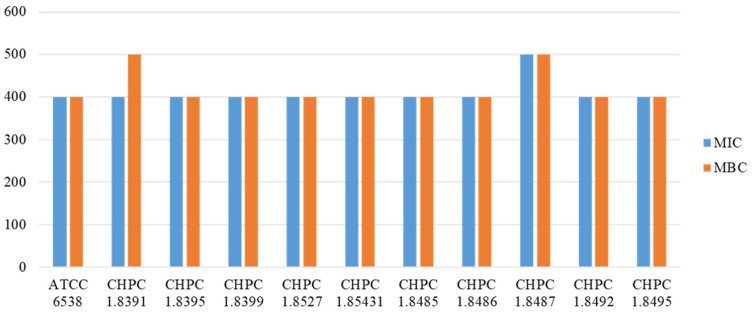
The MIC value and MBC value of 11 strains.

**Figure 4 microorganisms-11-00475-f004:**
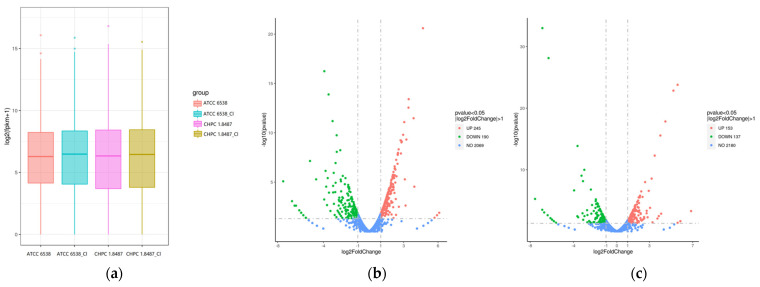
Gene expression analysis. (**a**) Box plot of the gene expression distribution. (**b**) Volcano plot of the genes differentially expressed in ATCC 6538. (**c**) Volcano plot of the genes differentially expressed in CHPC 1.8487. The red and green colors represent the upregulated and downregulated genes in (**b**,**c**), respectively.

**Figure 5 microorganisms-11-00475-f005:**
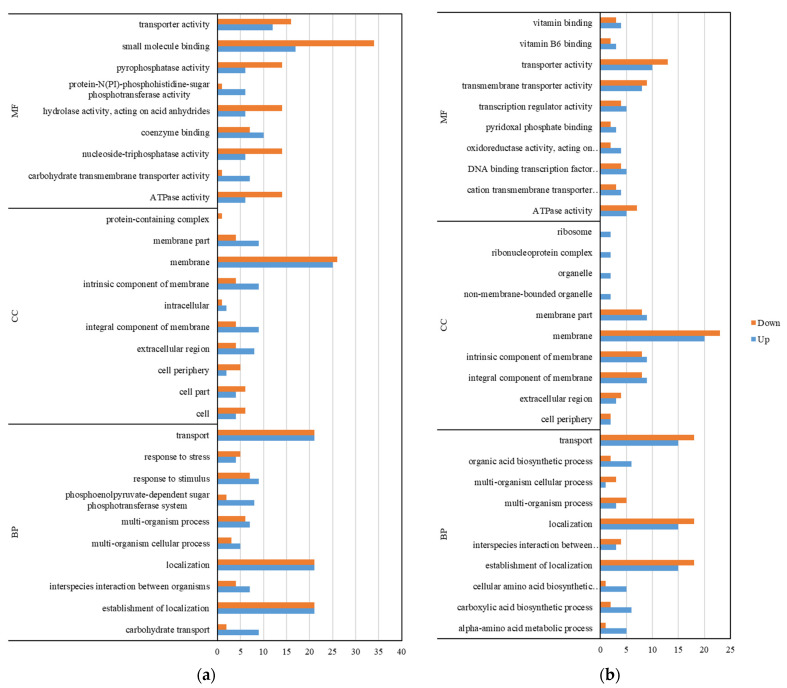
Gene Ontology (GO) functional annotation analysis of the differentially expressed genes (DEGs) with the treatment of chlorine-containing disinfectant. (**a**) GO functional annotation analysis of the DEGs in ATCC 6538. (**b**) GO functional annotation analysis of the DEGs in CHPC 1.8487.

**Figure 6 microorganisms-11-00475-f006:**
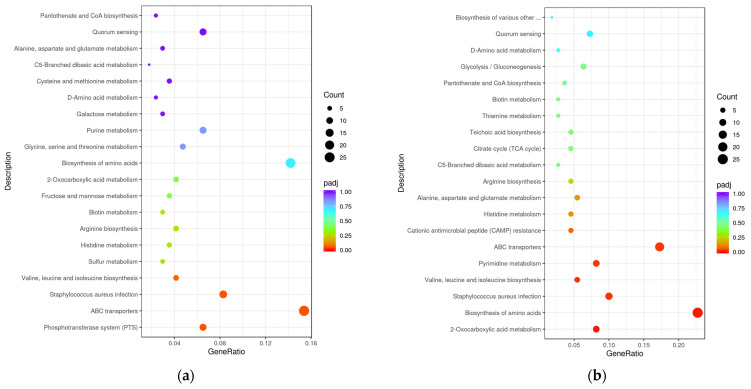
KEGG pathway enrichment analysis of DEGs with the treatment of chlorine-containing disinfectant. (**a**) KEGG pathway enrichment analysis of DEGs in ATCC 6538. (**b**) KEGG pathway enrichment analysis of DEGs in CHPC 1.8487. The bubble size indicates the number of genes, and the color shade represents the adjusted *p* value.

**Figure 7 microorganisms-11-00475-f007:**
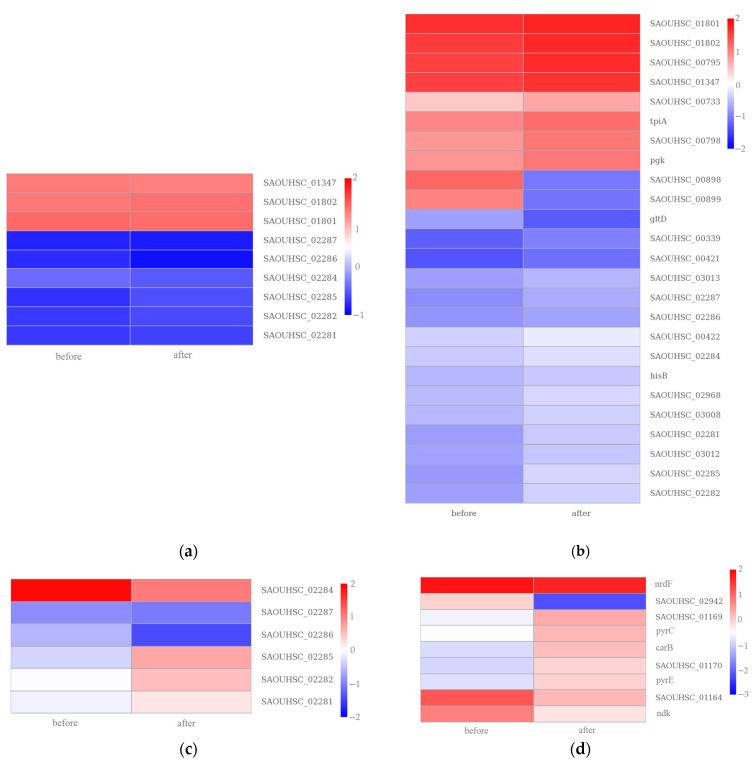
Transcriptional changes in the KEGG pathway related to (**a**) “2-oxocarboxylic acid metabolism”, (**b**) “biosynthesis of amino acids”, (**c**) “valine, leucine, and isoleucine biosynthesis”, and (**d**) “pyrimidine metabolism” in CHPC 1.8487 after the chlorine-containing disinfectant treatment.

**Table 1 microorganisms-11-00475-t001:** Functional classification of 14 strains of virulence factors predicted by VFDB.

VF Class	Virulence Factors	Related Genes
Adherence	Cell wall-associated fibronectin-binding protein	*ebh*
	Clumping factor A	*clfA*
	Clumping factor B	*clfB*
	Collagen adhesion	*cna*
	Extracellular adherence protein/MHC analogous protein	*eap/map*
	Intercellular adhesin	*icaD*
	Ser-Asp rich fibrinogen-binding proteins	*sdrC*
		*sdrD*
Enzyme	Serine protease	*splA*
		*splB*
		*splC*
		*splD*
		*splE*
		*splF*
Immune evasion	AdsA	*adsA*
	CHIPS	*chp*
	SCIN	*scn*
Secretion system	Type VII secretion system	*esaB*
		*esaD*
		*esaE*
		*esxB*
		*esxC*
		*esxD*
Toxin	Enterotoxin A	*sea*
	Enterotoxin B	*seb*
	Enterotoxin0like K	*selk*
	Enterotoxin0like Q	*selq*
	Exotoxin	*set*
	Leukotoxin D	*lukD*

**Table 2 microorganisms-11-00475-t002:** Neutralizer test results.

No.	Average Colony Count of Each Group (cfu/mL)						Error Rate among Groups 3, 4, and 5 (%)
	1	2	3	4	5	6	
1	0	18	22,300,000	24,700,000	25,300,000	0	4.98
2	0	9	16,800,000	17,000,000	18,700,000	0	4.57
3	12	180	25,800,000	27,500,000	29,100,000	0	2.19

**Table 3 microorganisms-11-00475-t003:** Quantitative germicidal test data.

Strain	Control Group		Experimental Group					
	Average Bacteria Concentration (cfu/mL)	N_0_	10 min			30 min		
			Average Bacteria Concentration (cfu/mL)	N_X_	KL	Average Bacteria Concentration (cfu/mL)	N_X_	KL
ATCC 6538	13,900,000	7.143	590	2.771	4.372	0	/	>7
CHPC 1.8391	15,000,000	7.176	1420	3.152	4.024	0	/	>7
CHPC 1.8395	11,900,000	7.076	410	2.613	4.463	0	/	>7
CHPC 1.8399	11,700,000	7.068	40	1.602	5.466	0	/	>7
CHPC 1.8427	16,300,000	7.212	110	2.041	5.171	0	/	>7
CHPC 1.8431	13,900,000	7.143	140	2.146	4.997	0	/	>7
CHPC 1.8485	14,000,000	7.146	0	/	>7	0	/	>7
CHPC 1.8486	13,800,000	7.140	30	1.477	5.663	0	/	>7
CHPC 1.8487	16,900,000	7.228	10	1.000	6.228	0	/	>7
CHPC 1.8492	12,400,000	7.094	30	1.477	5.617	0	/	>7
CHPC 1.8495	19,000,000	7.279	0	/	>7	0	/	>7

## Data Availability

The NCBI BioSampleObjects accession number of CHPC 1.8395 genome sequence information is SAMN33216256.

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
