# Peer review of "Screening of Staphylococcus aureus for Disinfection Evaluation and Transcriptome Analysis of High Tolerance to Chlorine-Containing Disinfectants"

_microorganisms, 2023, doi:10.3390/microorganisms11020475_

Round 1

Reviewer 1 Report (Previous Reviewer 2)

Dear Editor, Dear Authors,

The authors addressed my previous comments in this revised version.

regards

Q1. Problems about MIC experiment design. My concern is about the MIC testing performed. Indeed, classical MIC testing involve all to use initial bacterial concentration of around 10e5 bacteria/ml. Eucast is one of this highly standardized MIC determination protocole (https://www.eucast.org/ast_of_bacteria/mic_determination) Here the authors used the following protocole: “We adjusted the concentration of bacterial suspension to 1×10e8cfu/ml~5×10e8cfu/ml and used the distilled water to double dilute 84 disinfectants into test solutions of different concentrations (2000mg/L, 1000mg/L, 500mg/L). According to EN 1276 and Technical Standard For disinfection [24, 26], took 2.5ml of test solutions of various dilutions and added into the test tube containing 2.5ml nutrient broth (NB, Hopebio, Qingdao, China) of double concentration. We took 0.1ml of bacterial suspension with a bacterial content of about 108 cfu/ml into the test tube of NB containing disinfectant as the sample of the experimental group before inoculating the bacteria into the NB tube without disinfectant as the positive control sample.” If I understand clearly, the authors used 10e7/ml bacteria, much more than the classical concentrations used in almost all MIC determination protocoles. I am asking the authors to explain why using such protocole that is clearly not classical for MIC determination. 2 Response: Thank you for your suggestion. In the experiment, we took 0.1ml from the bacterial suspension with the concentration of about 10e8cfu/ml and added it into 5ml medium mixture of the prepared disinfectant and culture, and the actual action concentration was 5×10e5cfu/ml-5×10e6cfu/ml. For example, suppose the initial concentration is 10e8cfu/ml, and the calculation step of actual action concentration is (1×10e8cfu/ml×0.1ml)/5ml=2×10e6cfu/ml. We checked the documents of Eucast and found that it was applicable to drug sensitivity test with broth-microdilution method. In fact, at first, we planned to use this method for MIC experiment. But considering the research results will be applied to the disinfection experiment in China in the later stage, we changed to the nutrient broth dilution method from Technical Standard for Disinfection. Q2. Question about 10e9cfu/ml. Also in Figure 1, it looks that authors go to 10e9/ml bacteria. I am quite surprised by such a high bacterial count as S aureus reachs 10e8 bacteria/ml at OD of 1. How can the authors reached 10-times more bacteria than an OD of 1? Please check if calculations are correct. Response: We thank the reviewer for reading our paper carefully. At first, we used the OD600 value to calculate tthe bacterial concentration, but we could not determine that all of them were live bacteria. Later, we used the viable count method. The bacterial concentration of 10e9cfu/ml can only be produced in the optimal environment of Staphylococcus aureus when cultured at 37℃ for 24h in the medium with pH=7. This experiment has been repeated for three times, and the results were all about 10e9cfu/ml. The viable count of optimal environment, as a control, was significantly different from that in other environments. Q3. About Figure 3. About Figure 3.: having an MIC value multiplied by two-fold is not for me the sign of a resistance. MIC values classically can change, for a same antibiotics/disinfectant, by a factor 2. I do not believe going from 400 mg/L (for all strains) to 500 mg/L for strain supposed to resist the treatment is clearly for me not the sign of resistance to the treatment. The authors concluded “Therefore, it is suggested that CHPC 1.8487 has a relatively high resistance to chlorine-containing disinfectants”. I do not agree at all. Going from 400 to 500 mg/L is not the sign of a statistical resistance. Did the authors performed many time the MIC testing. In my hand, when doing n=3-6 MIC testing of antibiotics, it is often found a 2-fold variation in MIC value for the same molecule on the same strain. Response: 3 Thank you very much for your suggestion. We described the experimental process in more detail, in Lines 291-397 (Lines 346-352 in review mode) and Lines 421-427 (Lines 496-502 in review mode). At first, when the experimental concentration of chlorine-containing disinfectant is 1000mg/L, 500mg/L and 250mg/L, the MIC value and MBC value of all strains were 500mg/L. This result which was obtained by double dilution method is clearly. It showed that the experimental strains could not survive at 500mg/L effective chlorine, and could survive at 250mg/L. At this time, all candidate strains have the same long-term bacteriostatic effect on chlorine-containing disinfectant. Later, we wanted to explore the difference of each candidate strain, so we narrowed the range of disinfectant concentration, set 300mg/L and 400mg/L, and calculated trimmed mean through multiple experiments. In the process of reducing the range of MIC value, we found that the MIC value is not always limited to a small range. Therefore, we repeated the experiment 6 times and obtained the results of this study through trimmed mean. On this premise, CHPC 1.8487 has a higher chlorine resistance than other strains. We took it as the research object for preliminary exploration and analysis.

Reviewer 2 Report (Previous Reviewer 1)

Most of my concerns were addressed by authors in this new version of the manuscript.

This manuscript is a resubmission of an earlier submission. The following is a list of the peer review reports and author responses from that submission.

Round 1

Reviewer 1 Report

In this manuscript, by analyzing the resistance to chlorine of a collection of S. aureus strains, Li et al. found a strain that shows tolerance/resistance to this disinfectant. Then, the authors compare the transcriptome profile in the presence of chlorine between the resistant strain and a sensible strain to chlorine and identify probable genes/pathways that could be involved in the resistance to this disinfectant. Resistance to disinfectants is linked to antibiotic resistance, which represent a major threat for public health worldwide. In this sense, findings from the study by Li et al. sum to the knowledge about the bacterial responses to resist chlorine, a disinfectant commonly used.

The manuscript needs a carefully review of written.

Comments.

-At beginning, the manuscript seems to be focused on the analysis of disinfectant resistance on a collection of S. aureus strains; however, results start with an analysis of resistance to physical conditions and with the prediction of genetic determinants for virulence. I did not understand why these first analyses. What is the correlation of these results with those of disinfectant resistance? I suggest to start with the evaluation of disinfectant resistance.

-I did not understand the results of the Neutralizer test. Sorry. Explain little more the reasoning of this analysis.

-Lines 279-291. Describe results in a more simply way.

-Lines 295-301. Explain a little more the reasoning of this analysis. What concentration of disinfectant was tested?

-Genome comparative analysis could be useful to identify probable determinants specific for chlorine tolerance, present in the strain CHPC 1.8487 but not in the other S. aureus strains. Together with the results from this analysis you could describe the different resistance genes present in all strains assessed (Fig. 2).

Minor comments.

Line 60. Escherichia coli. Italics.

Lines 61 and 62. Pseudomonas aeruginosa. Italics.   

Lines 66-68. Rephrase sentence. Evaluate/ting/tion, repeated.

Line 156. Correct: 1x108, 5x108

Line 244. Name of genes. Italics.

Reviewer 2 Report

Dear Editor, Dear Authors,

I was invited to evaluate the following manuscript « Screening of Staphylococcus aureus for disinfection evaluation and transcriptome analysis of high tolerance to chlorine-containing disinfectants » by Li et al.

In their study, the authors evaluated the sensitivity of various strains of S aureus selected from the National Pathogen Resource Center of China to chlorine-containing disinfectants. For that, the authors measured the bacterial killing in different conditions : i) physical properties (acid resistance, alkaline resistance and heat resistance of strains), ii) genome-based safety evaluation (anal-ysis of virulence-related factors, drug resistance genes and disinfectant resistance genes) and iii) disinfection test evaluation (Minimum inhibitory concentration test, minimal bactericide concentration test and quantitative germicidal test for chlorine-containing disinfectant). The authors found one S. aureus strain highly resistant to chlorine-containing disinfectants which did not contain the mecA and the qac family genes. The authors then performed transcriptome analysis in this strain and compared it to the ones of the standard strain ATCC 6538 in the presence of low concentrations of chlorine-containing disinfectant. Data shown significant differential expression of the oxocarboxylic acid, amino acid and pyrimidine metabolisms allowing S. aureus to adapt to chlorine-containing disinfectants. Conclusions of the authors are that they established a technical process for screening and evaluating standard strains for disinfection, and identify bacterial evolution mechanism toward chlorine tolerance.

Overall, I found the study interesting but the experimental design has serious issues.

My concern is about the MIC testing performed.

Indeed, classical MIC testing involve all to use initial bacterial concentration of around 10e5 bacteria/ml. Eucast is one of this highly standardized MIC determination protocole (https://www.eucast.org/ast_of_bacteria/mic_determination)

Here the authors used the following protocole :

« We adjusted the concentration of bacterial suspension to 1×108cfu/ml~5×108cfu/ml and used the distilled water to double dilute 84 disinfectants into test solutions of different concentrations (2000mg/L, 1000mg/L, 500mg/L). According to EN 1276 and “Technical Standard For disinfection” [24, 26], took 2.5ml of test solutions of various dilutions and added into the test tube containing 2.5ml nutrient broth (NB, Hopebio, Qingdao, China) of double concentration. We took 0.1ml of bacterial suspension with a bacterial content of about 10e8cfu/ml into the test tube of NB containing disinfectant as the sample of the experimental group before inoculating the bacteria into the NB tube without disinfectant as the positive control sample. »

If I understand clearly, the authors used 10e7/ml bacteria, much more than the classical concentrations used in almost all MIC determination protocoles.

I am asking the authors to explain why using such protocole that is clearly not classical for MIC determination.

Also in Figure 1, it looks that authors go to 10e9/ml bacteria. I am quite surprised by such a high bacterial count as S aureus reachs 10e8 bacteria/ml at OD of 1. How can the authors reached 10-times more bacteria than an OD of 1 ?? Please check if calculations are correct.

About Figure 3 : having an MIC value multiplied by two-fold is not for me the sign of a resistance. MIC values classically can change, for a same antibiotics/disinfectant, by a factor 2. I do not believe going from 400 mg/L (for all strains) to 500 mg/L for strain supposed to resist the treatment is clearly for me not the sign of resistance to the treatment. The authors concluded « Therefore, it is suggested that CHPC 1.8487 has a relatively high resistance to chlorine-containing disinfectants ». I do not agree at all. Going from 400 to 500 mg/L is not the sign of a statistical resistance. Did the authors performed many time the MIC testing. In my hand, when doing n=3-6 MIC testing of antibiotics, it is often found a 2-fold variation in MIC value for the same molecule on the same strain.

Regards